# The Role of Moral Distress on Physician Burnout during COVID-19

**DOI:** 10.3390/ijerph19106066

**Published:** 2022-05-17

**Authors:** Caitlin A. J. Powell, John P. Butler

**Affiliations:** 1Department of Psychology, Thomas More University, Crestview Hills, KY 41017, USA; 2Department of Behavioral and Movement Sciences, Vrije Universiteit Amsterdam, 1081 HV Amsterdam, The Netherlands; j.p.butler@student.vu.nl

**Keywords:** burnout, moral distress, COVID-19, perceived organizational support

## Abstract

The purpose of this study was to explore the role of moral distress on physician burnout during COVID-19. Physicians in the US were interviewed between February and March 2021; 479 responded to our survey. The results indicated that moral distress was a key mediator in explaining the relationship between perceived organizational support, medical specialization, emotional labor, and coping with burnout. Results did not support increased burnout among female physicians, and contracting COVID-19 likewise did not play a role in burnout. Our findings suggest that physician burnout can be mitigated by increasing perceived organizational support; likewise, physicians who engaged in deep emotional labor and problem-focused coping tended to fare better when it came to feelings of moral distress and subsequent burnout.

## 1. Introduction

In late 2019, a novel coronavirus (COVID-19) was first identified and has since swept the globe. As health care workers and medical professionals are bearing the brunt of the burden in fighting this pandemic [1], this study sought to examine the mental health outcomes of physicians on the front line. Specifically, we explored the mediating role of moral distress [2] on physician burnout [3]. We also examined the roles of perceived organizational support [4] and emotional labor [5], as well as other factors, such as medical specialization and gender [6], to develop a clearer picture of physician outcomes during an ongoing pandemic and a stressed medical environment.

Burnout can be defined as a work-related syndrome and is assessed through the subcategories of emotional exhaustion, depersonalization (i.e., feeling detached from patients), cynicism, and a reduced personal sense of accomplishment [3,5]. Burnout is well documented in the medical profession and has been shown to lead to attrition in providers, as well as a deterioration in the quality of care [3,6]. However, though burnout has been well recognized in the medical community, less attention has been paid to physician-specific outcomes, as they are less often recognized as a vulnerable population within the medical field when it comes to morbidity and mortality outcomes [6]. This presents a clear gap in the literature; a search for scholarly articles on Google Scholar as of the time of publication showed that “physician” and “burnout” (138,000) generated less than half the results as “nursing” and “burnout” (460,000); EBSCOhost Academic Search Complete likewise revealed substantially fewer hits for physician burnout (5624 vs. 8025 results). Physicians face a demanding and stressful profession associated with high consequences and difficult decision making [6]. Research indicates that due to these working conditions, physicians are considered prone to burnout in addition to other occupational hazards such as suicidal ideation, insomnia, substance abuse, and PTSD [3,7].

These negative outcomes have been given new light as the COVID-19 pandemic has added greatly to the already significant levels of stress faced by health care professionals. Across the board, physicians have faced sharp increases in workload, a stark lack of resources (especially at the start of the pandemic), and an increased risk of becoming infected themselves [1,6], subsequently putting their families and loved ones at higher risk. An investigative report found that approximately 613 physicians in the US died from COVID-19 by April 2021 [8], and prior to vaccines being made available, approximately 14% of COVID-19 cases were health care personnel [9]. While physicians were found to have a lower risk of hospitalization compared to nursing-related occupations [10,11], patient-facing health care professionals tended to face generally higher risk [12]. In addition, health care workers experienced increases in workplace violence, including verbal assault, threats, harassment, and ostracization [13]. In addition, physicians faced a lack of sleep and increased workload, which directly contributed to burnout [6].

One of the key factors that could have contributed to burnout among physicians is moral distress. Moral distress is the discomfort or internal conflict that is caused when professionals feel as though they cannot carry out the appropriate or ethical course of action that they believe to be right [2,14,15,16]. More specifically, it is the experience of psychological distress following the experience of a moral event, such as the withdrawal of life-sustaining treatments [17]. Moral distress has been studied mostly among military and first responders and has been shown to increase burnout and attrition among that population [15,16], although recent research has expanded the study of moral distress to recognize that it is experienced among a wide range of medical professionals [18]. Causes of moral distress are varied and have been shown to include powerlessness (at patient/family, team, and organizational levels), end-of-life issues, and poor team function [19]. The early days of the pandemic were characterized by a shortage of supplies and staff, often rapidly changing information about appropriate preventative measures [20,21], and regional surges in patient numbers leading to reduced bed space [22]. Due to the lack of resources, information, and staff that characterized the beginning of the COVID-19 pandemic, we hypothesize that this led to a perceived lack of ability for physicians to sufficiently deliver care, increasing their moral distress. Furthermore, in cases of futility, aggressive medical treatment at the end of life is a well-documented source of moral distress among critical care nurses [2,14,18]; this is often a marker of critical COVID-19 care [23]. Recent research has found increases in moral distress [24] and burnout [25] among health care workers who worked with COVID-19 patients. Therefore, we hypothesized that moral distress would contribute towards physician burnout during COVID-19.

Additionally, we sought to examine occupational factors previously linked to burnout. Specifically, we examined perceived organizational support and emotional labor as both these facets have been linked to burnout among health care workers and are resultant of organizational policies and approaches regarding employees [26,27].

Perceived organizational support is the willingness of an organization to reward work, satisfy social and emotional needs, and value employee well-being [4]. In the previous 2002 SARS outbreak, organizational support was demonstrated to reduce burnout as organizational support acted to lessen the impacts of emotional exhaustion by providing the necessary informational and emotional support and making employees feel well equipped, protected, and supported by their hospital [26]. Occupational factors have also been shown to contribute to physician burnout during the COVID-19 pandemic [6,28]. We expected an increase in stressors stemming from the organization’s policies may have accentuated the physicians’ perception that they cannot provide adequate care, therefore leading to an increase in moral distress in cases of lowered perceived organizational support.

Emotional labor is the act of expressing organizationally desired emotions during service transactions and requires the management of an employee’s actual felt emotions when there is a discrepancy between these emotions and the emotions the organization wishes them to display [5,27]. This aspect of emotional labor has become an ever-growing facet of the medical profession as hospitals have begun to emphasize not only medical outcomes but also the patient experience within the health care system [27]. We hypothesize that this can increase moral distress as this requirement to regulate emotions under duress may lead to greater levels of emotional exhaustion, a key marker of burnout [5].

Recent research has established increased burnout among female physicians during COVID-19 [6]; we were interested in exploring the possible role of emotional labor when it comes to female physician burnout. Previous work has demonstrated that female health care workers are often under greater pressure to perform emotional labor due to gender norms and expectations [29]—for example, research shows that female physicians spend nearly 16% more time with patients on average, often taking that time to engage in more communication and shared decision-making [30], which requires additional emotional labor. Therefore, we expect that a subsequent increase in emotional labor will act as a mediator in the relationship between gender and burnout.

Finally, we sought to examine other demographic factors such as the physician’s medical specialization and self or familial contact with COVID-19, as well as the role of coping methods in the relationship between moral distress and burnout.

Our hypotheses are as follows:

**Hypothesis** **1** **(H1).**
*There is a direct relationship between moral distress and burnout during the COVID-19 pandemic.*


**Hypothesis** **2** **(H2).**
*The relationship between perceived organizational support and burnout will be significantly mediated by moral distress.*


**Hypothesis** **3** **(H3).**
*The relationship between emotional labor and burnout will be significantly mediated by moral distress.*


**Hypothesis** **4** **(H4).**
*Female physicians and those who work in critical care will report higher levels of burnout due to increased emotional labor demands.*


**Hypothesis** **5** **(H5).**
*Those physicians that have contracted COVID-19 will experience greater moral distress and subsequent burnout.*


**Hypothesis** **6** **(H6).**
*Those physicians whose family members have contracted COVID-19 will experience greater moral distress and subsequent burnout.*


**Hypothesis** **7** **(H7).**
*Those physicians whose medical specialty is in emergency medicine will experience greater moral distress and subsequent burnout.*


**Hypothesis** **8** **(H8).**
*Those physicians who practice positive coping techniques will experience less burnout.*


## 2. Materials and Methods

### 2.1. Power Analysis

An a priori power analysis using joint significance in G-Power [31] found a sample size of 405 to be sufficient if anticipating a medium effect for mediation.

### 2.2. Participants

Participants were recruited online from physician Facebook groups, including a group consisting of physician parents and a group consisting of emergency medicine physicians. Physicians were recruited through a post on those social media platforms asking them to volunteer for a survey targeting physicians during the pandemic to ‘determine how and why particular factors such as organizational support and moral distress mediate burnout.’ Participants were informed that their results would be kept confidential and that the study had been IRB approved. Other than stating that the study was specifically recruiting physicians, we did not otherwise provide any restrictions on participation. Participants voluntarily completed the survey via SurveyMonkey, clicking on the survey link provided in the recruitment posts.

### 2.3. Procedure

After clicking the link on the recruitment post to the survey, participants completed an informed consent form and a demographic questionnaire asking for their gender identity, the state or states where they were licensed to practice, their age, and their area of medical specialization. After, they then completed questionnaires for burnout, moral distress, perceived organizational support, work motivation, emotional labor, and coping. They were then thanked for their participation.

Data were collected between 22 February 2021 and 29 March 2021. According to the CDC, by the end of March 2021, 559,637 deaths had been attributed to COVID-19 in the US. While the number of weekly COVID-19 deaths peaked in the week of 1 January 2020 at 25,685, during the time the data were collected, the average weekly COVID-19 deaths averaged 6858 [32]. These data were collected between the second and third ‘waves’ of COVID-19 in the US, approximately one year after widespread shutdowns in March 2020 [33].

Each original scale range was adjusted to fit a 7-point Likert type scale where 1 = “not at all true, and 7 = “absolutely true”. The individual scales are provided below.

The Oldenburg Burnout Inventory [34] consisted of 15 items and had a high reliability (*α* = 0.871). Example question: “After work I have enough time for leisure activities”.

The Moral Distress Scale [2] was a modified version of the moral distress in critical care nurses scale. Modifications made were with the intent to better target physicians rather than nurses. As an example: “Follow physicians’ request not to discuss code status with family when patient is incompetent” was changed to “Opt to not discuss code status with family when patient is incompetent”. This modified scale consisted of 20 items and had a high reliability, which was comparable to that of the original scale (our scale *α* = 0.884; original scale *α* = 0.930). Example question: “Do nothing when a colleague is, in my own opinion, providing incompetent care”.

The Perceived Organizational Support [35] scale consisted of 8 items and had a high reliability (*α* = 0.939). Example question: “The organization values my contribution to its well-being”

The Work Motivation [36] scale consisted of 19 items and had a sufficient reliability of *α* = 0.771. Example question: “Why do you put effort into your job… because others will respect me more”.

The Emotional Labor [29] scale was split into subscales, each of which was analyzed separately. The first subscale was “Surface”, consisting of 3 items, with a reliability of *α* = 0.825. Example question: “I put on an act in order to deal with patients/colleagues in an appropriate way”. The second subscale was “Deep”, consisting of 3 items, with a reliability of *α* = 0.557. Example question: “I try to actually experience the emotions that I must show to patients/colleagues”. The third subscale was “Natural Emotions”, consisting of 3 items, with a reliability of *α* = 0.802. Example item: “The emotions I express to patients/colleagues are genuine”. The fourth subscale was “Termination,” consisting of 3 items, with a reliability of *α* = 0.230. Example item: “When patients/colleagues disapprove of my service I will choose silence”. Due to not achieving the basic threshold of a reliability of 0.50 or higher, the “Termination” subscale was excluded from subsequent analyses.

We measured coping using the Brief COPE scale [37]. Based on previous research [38], the COPE scale was split into three sub-sections. The first sub-scale was Problem-Focused Coping and consisted of 6 items (*α* = 0.734). Example item: “I’ve been concentrating my efforts on doing something about the situation I’m in.” The second sub-scale was Emotional-Focused Coping and consisted of 11 items (*α* = 0.502). Example item: “I’ve been joking about my circumstances”. The last sub-scale was Avoidant-Focused Coping and consisted of 6 items (*α* = 0.465). Example item: “I’ve been refusing to believe that this happened.” Due to not achieving the basic threshold of a reliability of 0.50 or higher, the Avoidant-Focused Coping subscale was not included in subsequent analyses.

### 2.4. Statistical Methods

The data were analyzed as follows; first, a series of descriptive statistics were run on the demographic information, including mean and standard deviation age and years spent practicing, as well as frequency counts for regions in the US and medical specialization. Then, the descriptive statistics were run on each scale. A series of bivariate correlations determined the relationships between burnout and our independent variables of moral distress, perceived organizational support, work motivation, problem-focused coping, emotion-focused coping, ‘surface’ emotional labor, and ‘natural emotions’ emotional labor. Next, a series of *t*-tests were run to examine the difference in burnout based on physician gender, medical specialization, COVID-19 status, and the COVID-19 status of close family on burnout. Lastly, a series of mediations were conducted using a bootstrapping technique for measuring indirect and direct effects on variables [39].

## 3. Results

### 3.1. Demographics

The sample consisted of 479 physicians (105 Male, 373 Female, 1 Other) ranging in age from 27 to 66 (Mage = 41.65 ± 8.23). On average, the sample reported having practiced for 12.74 years ± 7.62. All those who completed the survey were physicians, and they represented various medical specializations ranging from private practice to emergency medicine. Most of the physicians were licensed to practice in the US, and 1% of the sample were non-US; the US-practicing physicians represented all five major regions (see Appendix A for complete demographic information).

### 3.2. Descriptive Statistics

Means and standard deviations for each scale/subscale are included in Table 1.

### 3.3. Correlations

Key results indicated that burnout was significantly positively correlated with emotion-focused coping, ‘surface’ emotional labor, and moral distress and was negatively correlated with problem-focused coping, ‘natural’ emotional labor, ‘deep’ emotional labor, and organizational support.

Other key findings included the negative correlation between emotion-focused coping and years of experience, as well as deep emotional labor and years of experience, and the positive correlation between emotion-focused coping and problem-focused coping. Workplace motivation was positively correlated with moral distress, problem-focused coping, and emotion-focused coping. In addition, ‘surface’ emotional labor was negatively correlated with ‘natural’ emotional labor (see Table 2 for all correlations).

### 3.4. T-Tests

The results indicated that emergency medicine specialists (*M* = 4.18 ± 0.59) reported higher burnout than those not in emergency medicine (*M* = 3.98 ± 0.52), *t*(315) = −2.65, *p* = 0.009, *d* = −0.366; there were no differences in burnout based on physician gender, whether participants reported being diagnosed with COVID-19, or whether participants reported close family being diagnosed with COVID-19 (see Table 3 for all relevant statistics).

### 3.5. Mediations

The mediational analyses supported our main hypothesis; we found that moral distress significantly mediated the relationship between perceived organizational support and burnout. Moral distress also mediated the relationship between EM vs. non-EM professionals and burnout, where those in emergency medicine specializations experienced increased moral distress and subsequent burnout. Our results also found that moral distress was a significant mediator for those who engaged in problem-focused coping, those who rated higher on deep emotional labor, and those who rated higher on natural emotional labor where higher ratings on these factors experienced less moral distress and subsequent burnout. The opposite was found for surface emotional labor and emotion-focused coping, such that higher ratings on those factors experienced more moral distress and burnout (see Table 4 for relevant statistics).

## 4. Discussion

We found support for hypotheses 1–3 and 7–8. Overall, our correlations showed clear links between moral distress and burnout, and moral distress was found to be a significant mediator across a wide range of professional and demographic factors, including perceived organizational support, various types of emotional labor and coping, and specializing in emergency medicine. In essence, a key driver of burnout among physicians during the first year of the pandemic was borne out of a notion that they were unable to perform to their own ethical standards; that they felt as though they could not sufficiently perform their work.

Being diagnosed with COVID-19 did not appear to have a substantial impact on our outcomes of interest; in our survey, only about 12% of our sample had contracted COVID-19, which may have contributed to a non-significant finding. One factor that is worth exploring in the future is the potential role of long COVID in physician burnout—results suggest that nearly one in four COVID-19 patients experienced lingering symptoms, with similar rates for those who were not hospitalized and those with mild or asymptomatic cases [40]. Long COVID would presumably impact physicians infected with COVID-19 at similar rates, and in anecdotal reports, physicians have reported struggling with long COVID, including persistent problems with fatigue and brain fog [41]; a sample focused solely on physicians who have contracted COVID-19 may be more illuminating.

We were surprised at our lack of significant findings around physician gender; previous research would suggest that female physicians tend to have higher burnout [42] and have had higher burnout during COVID-19 in particular [6]. We found no results for differences in burnout between male and female physicians, and, as a result, emotional labor did not mediate between the two factors. We also tested moral distress as a mediator, which was also non-significant; while previous research found that female critical care nurses had higher moral distress ratings than male critical care nurses [43], their sample size was small (*N* = 31) and potentially skewed by an outlier. Our sample was disproportionally female (approximately 77%); this is possibly due to our sampling methods, which could be a factor; future research should continue to explore whether gender differences are widely supported.

Results identified a few vital ways to address physician burnout, the most obvious being perceived organizational support. Increased support decreased moral distress and burnout, indicating the vital role organizations can play; this echoes recent calls for increases in organizational support as a way to address burnout [25]. In addition, those who had better perceived organizational support also rated higher in other factors found to be negatively associated with burnout, such as problem-focused coping, deep emotional labor, and natural emotional labor. The AACN’s Model to Rise Above Moral Distress emphasizes the four A’s: ask, affirm, assess, and act [44]. These results would suggest that people in supportive organizations may feel more empowered to ‘ask’ and ‘act’, and to engage in problem-focused coping. Moral distress likewise played a role in these factors; those who engaged in problem-focused coping, deep emotional labor, and natural emotional labor experienced less moral distress and less subsequent burnout than those who engaged in emotion-focused coping and surface-level emotional labor. This corresponds to previous research emphasizing the importance of alignment between empathy and perspective taking (among other factors) in navigating morally distressing situations [45]. Years spent practicing was paradoxically negatively related to both emotion-focused coping (associated with greater moral distress and burnout) and deep emotional labor (associated with less moral distress and burnout). This would suggest that physicians who have had more experience may be more likely to emotionally disengage—both in terms of coping and in interactions with their patients—which can have both benefits and detriments when it comes to burnout.

It is necessary to contextualize our findings; results were collected in February and March of 2021, approximately a year after the US experienced widespread shutdowns due to COVID-19. Vaccines had been available for 7 months at the time of our survey administration; by the end of our data collection period, 15% of the US population had been fully vaccinated (approximately 51 million). While vaccine access differed from state to state, they were mostly restricted to the elderly, high risk, and front-line workers [46]. Reports found that physician vaccination rates at the time of survey administration were around 75%, leading in health care worker vaccination rates [47]. While this lent a certain degree of increased protection to physicians at the time of survey collection, the impact of the previous year certainly continued to be a factor.

It is likewise important to provide study limitations; as stated above, this study presents a snapshot of physicians during a specific time period in the pandemic; as a result, our findings are limited in terms of generalizability. In addition, we utilized convenience sampling; while our sample contained physicians from a wide range of specializations across the US, we would practice caution in broad generalizations. Lastly, we did not collect data on race/ethnicity.

## 5. Conclusions

As of the time of this manuscript submission, COVID-19 is well into its second year, and the lingering impacts of the pandemic will continue to be felt for years to come. This research contributes to a potentially less well-studied topic in the experience of physicians during COVID-19, namely, the role of moral distress. As discussed in the introduction, moral distress is typically studied among trauma nurses in previous research. Due to the unique circumstances of the pandemic, which brought with it supply-chain, staffing, and treatment deficits, we believe that moral distress among physicians during COVID-19 is a particularly apt area of study. This research supports the key role moral distress plays in understanding physician burnout during COVID-19.

## Figures and Tables

**Table 1 ijerph-19-06066-t001:** Means and standard deviations of scales.

Scale	Mean ± SD
OBI (burnout)	4.14 ± 0.57
Perceived Organizational Support	2.90 ± 0.97
Moral Distress	3.86 ± 0.57
Workplace Motivation	1.88 ± 0.44
Surface Emotional Labor	2.86 ± 1.02
Deep Emotional Labor	3.52 ± 0.68
Natural Emotional Labor	3.53 ± 0.80
Problem-Based Coping	3.52 ± 0.61
Emotion-Based Coping	3.32 ± 0.43

**Table 2 ijerph-19-06066-t002:** Correlations between measures, years of practice.

	1	2	3	4	5	6	7	8	9	10
1. POS	1	−0.193 **	−0.245 **	0.235 **	0.240 **	0.226 **	−0.065	0.018	−0.456 **	−0.025
2. MD		1	0.191 **	−0.167 **	−0.226 **	−0.027	0.063	−0.056	0.187 **	0.163 **
3. SEL			1	−0.305 **	−0.725 **	−0.083	0.117 *	−0.064	0.472 **	0.059
4. DEL				1	0.395 **	0.257 **	0.097	−0.107 *	−0.266 **	0.094
5. NEL					1	0.123 *	−0.047	0.035	−0.399 **	0.003
6. PBC						1	0.515 **	0.024	−0.140 *	0.281 **
7. EBC							1	0.185 **	0.139 *	0.321 **
8. Years								1	−0.009	−0.046
9. OBI									1	−0.041
10. WM										1

POS = perceived organizational support; MD = moral distress; SEL = surface emotional labor; DEL = deep emotional labor; NEL = Natural emotional labor; PBC = problem-based coping; EBC = emotion-based coping; Years = years of practice; OBI = Burnout; WM = workplace motivation; * significant at *p* ≤ 0.05, ** significant at *p* ≤ 0.01.

**Table 3 ijerph-19-06066-t003:** *T*-tests exploring differences in burnout.

Groups	Group 1 M ± SD	Group 2 M ± SD	*T*	*Df*	*p*	*d*
EM vs. Non-EM *	4.18 ± 0.59	3.97 ± 0.52	2.65	315	0.009 **	0.366
Male vs. Female	4.14 ± 0.58	4.14 ± 0.57	−0.01	355	0.989	−0.002
COVID-19 vs. No COVID-19	4.15 ± 0.57	4.14 ± 0.57	0.07	355	0.947	0.011
Family COVID-19 vs. No COVID-19	4.12 ± 0.58	4.15 ± 0.57	−0.41	356	0.684	−0.044

* EM refers to emergency medicine; ** significant at *p* ≤ 0.01

**Table 4 ijerph-19-06066-t004:** Impact of moral distress, emotional labor (EL), and coping on physician burnout.

IV	Mediator	DV	Point Estimate	SE	LLCI	ULCI
Gender	Moral distress	burnout	0.001	0.012	−0.023	0.026
Gender	Surface EL	burnout	−0.017	0.068	−0.154	0.115
Gender	Deep EL	burnout	0.007	0.026	−0.045	0.062
Gender	Natural EL	burnout	0.012	0.045	−0.077	0.101
Years practiced	Problem-focused coping	burnout	<0.001	0.001	−0.001	0.002
Years practiced	Moral distress	burnout	<0.001	0.001	−0.001	0.001
EM	Moral distress	burnout	−0.024	0.013	−0.053	−0.032
POS	Moral distress	burnout	−0.028	0.012	−0.052	−0.003
Deep EL	Moral distress	burnout	−0.028	0.011	−0.053	−0.009 *
Natural EL	Moral distress	burnout	−0.028	0.012	−0.053	−0.005 *
Surface EL	Moral distress	burnout	0.028	0.012	0.006	0.053 *
Problem-focused coping	Moral distress	burnout	−0.018	0.010	−0.039	−0.001 *
Emotion-focused coping	Moral distress	burnout	0.027	0.014	0.005	0.058 *

POS refers to perceived organizational support; EL refers to emotional labor. Confidence intervals bias-corrected and accelerated; 1000 bootstrap samples; confidence set at at 95%; includes correction for median bias and skew. * significant at *p* ≤ 0.05.

## Data Availability

Data and raw questionnaires are available through the Open Science Framework at https://osf.io/a3btm/ (accessed on 16 May 2022).

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
