# Peer review of "The Role of Moral Distress on Physician Burnout during COVID-19"

_ijerph, 2022, doi:10.3390/ijerph19106066_

Round 1
Reviewer 1 Report
Thank you for allowing me to review this very interesting and relevant manuscript. Whilst there has been much written around burnout and to a lesser extent around moral distress in physicians during the covid-19 pandemic, including the importance of perceived organizational support as a mediating factor, this study, I believe, adds an additional dimension to this by looking at emotional labour and coping strategies and as such, I feel that with some minor modification, it merits publication.
I have a few minor comments which I wondered if the authors would like to consider.
- Abstract: line 12 - I am unsure of the use of the phrase "there was no support for increased burnout...". An alternative may be "Results did not support increased burnout...".
- Introduction: lines 34 - could the authors expand more on what they mean by the phrase "less attention has been paid to physician specific outcomes..."? What "outcomes" are being referred to? I am also not sure what "gap in the literature" exists that the authors have identified which merits further study as it is not clear in the text . An additional explanation to clarify this, with a further citation to support this claim, would be worthwhile.
- Introduction: lines 67-68 - "we hypothesized that moral distress would be a marker of physician burnout during COVID-19". Moral distress is usually an acute emotional state experienced by a healthcare worker under certain circumstances whereas burnout is a syndrome brought on by an accumulation of factors over a period of time. Thus I am unsure whether it is correct to hypothesize that one is a "marker" of the other as they both measure different constructs. I suspect it is more likely the former will be a significant contributing cause for the latter through time but as we know, burnout also exists for other reasons in the absence of moral distress.
- Conclusions: Lines 292-293 - "This research provides a potentially unexplored topic in the experience of physicians during COVID-19; namely, the role of moral distress". It may be more accurate to refer to this as a "less well studied topic" as other studies have more recently been published which have explored this theme (e.g. Spilg EG et al. The new frontline: exploring the links between moral distress, moral resilience and mental health in healthcare workers during the COVID-19 pandemic. BMC Psychiatry. 2022 Dec;22(1):19. https://doi.org/10.1186/s12888-021-03637-w).
Reviewer 2 Report
The paper is interesting, and the methodology of the study is clear and appropriate for this type of research.
The introduction does not provide sufficient background, since it does not reflect all the previous research about moral distress in Nursing. On line 57, the authors state "Moral distress has been studied mostly among military and first responders, and has been shown to increase burnout among that population". The authors should mention some of the relevant work about moral distress in Nursing field. These are only some examples of relevant books and papers:
https://link.springer.com/book/10.1007/978-3-319-64626-8
https://www.ncbi.nlm.nih.gov/pmc/articles/PMC6175312/
https://ojin.nursingworld.org/MainMenuCategories/EthicsStandards/Resources/Courage-and-Distress/Understanding-Moral-Distress.html
https://pubmed.ncbi.nlm.nih.gov/33888021/
https://pubmed.ncbi.nlm.nih.gov/16767017/
https://pubmed.ncbi.nlm.nih.gov/23777328/
In addition, the methodology section should be described better, with more well-organized details.
Reviewer 3 Report
The manuscript is all-encompassing and full-bodied .. I have concerns about the structure of the manuscript (reshaping the statistical analysis paragraph) and the lack of bibliographic references to discuss
21 is not the correct acronym
28 references missing
105 H:1 …
113 Loved one contract?
122 As this is the methods section, no results can be presented. Of the participants it is necessary to describe, how and when they were selected, not how many. Were they all doctors? other professional class? Were there any age cut-offs? Did the design collect from the first lockdown? passed the first wave?
180 Statistical Analysis totally missing..
It is tedious to defer any data to supplemental materials
Reshape the results, please. the beginning of the sub-paragraphs are statistical methods to be reported in the appropriate section in the methods. Represent by convention mean±SD
235 Inelegant to start the discussion like this .. Provide the hypotheses in full, to make the reader more readable
258 precisely, define them better in the methods
260 I can suggest on this topic the references to the current literature referring to the impact of boards, managerial figures of organizations and their management and reliability in that context. For example: https://pubmed.ncbi.nlm.nih.gov/33801349/ ; https://pubmed.ncbi.nlm.nih.gov/34574600/
